# Platelet Function Disturbance During Veno-Venous ECMO in ARDS Patients Assessed by Multiple Electrode Aggregometry—A Prospective, Observational Cohort Study

**DOI:** 10.3390/jcm8071056

**Published:** 2019-07-19

**Authors:** Saskia Wand, Jan Felix Huber-Petersen, Joern Schaeper, Claudia Binder, Onnen Moerer

**Affiliations:** 1Department of Anesthesiology, University Medical Center Goettingen, 37075 Goettingen, Germany; 2Department of Anesthesiology, Ruhr-University Bochum, St. Josef- and St. Elisabeth Hospital, 44791 Bochum, Germany; 3Department of Thoracic, Cardiac and Vascular Surgery, University Medical Center Goettingen, 37075 Goettingen, Germany; 4Department of Hematology and Oncology, University Medical Center Goettingen, 37075 Goettingen, Germany

**Keywords:** extracorporeal circulation, ECMO, multiplate, platelet function, multiple electrode aggregometry

## Abstract

Extracorporeal (veno-venous) membrane oxygenation (vvECMO) has been shown to have negative effects on platelet number and function. This study aimed to gain more information about the impact of vvECMO on platelet function assessed by multiple electrode aggregometry (MEA). Twenty patients with the indication for vvECMO were included. Platelet function was analyzed using MEA (Multiplate^®^) before (T-1), 6 h (T0), one (T1), two (T2), three (T3), and seven (T4) days after the beginning of vvECMO. Median aggregational measurements were already below the normal reference range before vvECMO initiation. Platelet aggregation was significantly reduced 6 h after vvECMO initiation compared to T-1 and spontaneously recovered with a significant increase at T2. Platelet count dropped significantly between T-1 and T0 and continuously decreased between T0 and T4. At T4, ADP-induced platelet aggregation showed an inverse correlation with the paO2 in the oxygenator. Platelet function should be assessed by MEA before the initiation of extracorporeal circulation. Although ECMO therapy led to a further decrease in platelet aggregation after 6 h, all measurements had recovered to baseline on day two. This implies that MEA as a whole blood method might not adequately reflect the changes in platelet function in the later stages of extracorporeal circulation.

## 1. Introduction

Veno-venous extracorporeal membrane oxygenation (vvECMO) is an increasingly used therapeutic option in patients with severe acute respiratory distress syndrome (ARDS), when conventional management strategies, such as adequate PEEP-setting or prone positioning, fail to secure a sufficient gas exchange [1,2,3].

The concerns to apply vvECMO for less severe states of ARDS despite its benefits, e.g., in order to enhance lung protective ventilation, results from inherent and substantial complications related to the technique, which prevail despite the technical advances related to the technique of extracorporeal membrane oxygenation [2,3]. 

ECMO therapy still imposes massive changes to the body’s normal physiology and entails the risk for severe complications [1,4,5]. In particular, dysfunction of the coagulation system with consecutive thromboembolic or hemorrhagic events are observed frequently [6,7,8].

Platelets are an important factor for an efficient hemostasis. Extracorporeal circulation in cardiopulmonary bypass as well as in ECMO therapy has been shown to have negative effects on platelet number and function [7,9]. In recent years, the understanding has grown that monitoring of platelet function is much more important for the evaluation of a bleeding event than the mere analysis of platelet number. The availability of multiple electrode aggregometry (MEA) as a point-of-care device for the bedside evaluation of platelet function from whole blood has led to its widespread use and to the implementation of MEA in algorithm-based treatment of hemorrhage [10,11]. However, it is unclear to which extent results from studies using MEA to evaluate the effects of short-term extracorporeal circulation, such as cardiopulmonary bypass, on platelet function, can be extrapolated on the effects during long-term extracorporeal circulation as in vvECMO therapy [9,12]. Studies in animal models suggest that there may be an additional effect of the severe underlying disease in ARDS on platelet dysfunction assessed with MEA [13]. 

The aim of this study was to gain more information about the impact of vvECMO therapy on platelet function using MEA.

Part of the study data has been presented as an abstract on national and international congresses [14] and has been submitted elsewhere.

## 2. Experimental Section

### 2.1. Trial Design and Patients

This observational, prospective monocentric cohort study was conducted at the University Medical Center Goettingen, Germany. The present study complies with the declaration of Helsinki and was approved by the local Scientific and Ethics Review Board on 4 December 2013 (local reference number: 19/5/13). The study was retrospectively registered at ClinicalTrials.gov (NCT03754868). 

### 2.2. Patient Enrollment

After written informed consent was obtained, 20 consecutive patients with the indication for vvECMO therapy were enrolled in the study. As none of the patients was conscious at the time of enrollment, consent was primarily obtained from the appointed legal guardian. Due to the fact that indication for ECMO was not to be delayed, it was allowed to include the patient based on the presumed will assessed via the treating clinician if the relatives could not be accessed, and official consent/denial was obtained thereafter. In case of refusal, data collection was stopped, and the patient was excluded from the study. Written informed consent was obtained from the patients retrospectively in the case of full recovery. Patients were not eligible for the study if they were below 18 years of age, suffered from a known coagulation disorder, or were pregnant. 

The indication for vvECMO treatment was set by the responsible physician of the ECMO team. After the ECMO/ARDS team had been contacted for support, patients were either placed on ECMO in the referral hospital and retrieved on ECMO or transported to the ARDS center and eventually placed on ECMO, if conservative treatment failed.

### 2.3. ECMO Setup

vvECMO was established using either a ROTAFLOW^®^ (Maquet Critical Care, Gettinge Group, Solna, Sweden) centrifugal pump or CARDIOHELP^®^ system (Maquet Critical Care) equipped with a Quadrox-D^®^ oxygenator (Maquet) and the Maquet Permanent Life Support (PLS) system, respectively connected via a bio-coated tubing set (Maquet Critical Care). Venous access was established percutaneously using the right femoral and vein for blood outflow (21–25 Fr lumen HLS Cannulae, Maquet Critical Care), and the right internal jugular vein (17–23 Fr) for return flow. If femoral-jugular access was impossible, either bi-femoral or single venous access, using a double lumen cannula (32 Fr Avalon Elite Bi-Caval^®^), was chosen. 

### 2.4. Hematological/Hemostaseological Data Collection and Time Points

Blood samples for conventional laboratory parameters including hemoglobin, leukocyte count, platelet count, fibrinogen, prothrombin time (PT), and activated partial thromboplastin time (aPTT) were drawn daily. Additionally, blood samples for platelet aggregometry were collected before (T-1), 6 h (T0), 1 day (T1), 2 days (T2), 3 days (T3), and 7 days (T4) after the start of ECMO therapy.

### 2.5. Multiple Electrode Aggregometry (MEA)

MEA was performed using the whole blood impedance aggregometer Multiplate^®^ (Roche AG, Grenzach, Germany). MEA results were considered valid if the platelet count was above a selected cut-off of 70.000/µL. The methodology of MEA has been previously described elsewhere [15,16]. Blood samples were collected into heparin-anticoagulated and calcium-balanced tubes (Bloodgas-Monovette, Sarstedt AG, Nümbrecht, Germany). Platelet aggregation was initiated by stimulation with arachidonic acid (AA) (0.5 mmol/L; ASPItest), adenosine diphosphate (ADP) (6.4 µmol/L; ADPtest) and thrombin receptor activating peptide 6 (TRAP-6) (32 mmol/L; TRAPtest). Platelet aggregability was calculated as area under the aggregation curve (AUC), which was presented in arbitrary “aggregation units” (AU). Reference ranges given by the manufacturer are 79–141 AU for AA-induced, 55–117 AU for ADP-induced, and 92–151 AU for TRAP-6-induced platelet aggregation. Quality controls for the aggregometer were performed regularly according to the manufacturer’s instructions.

### 2.6. Sample Size and Statistical Analyses

The presented study had an explorative nature and no data was available regarding the effects of ECMO therapy on MEA measured with Multiplate^®^ at the time of planning. Therefore, no sample size calculation was performed. Depending on the distribution of the data from the Kolmogorov–Smirnov test, the results are given as the mean ± standard deviation (SD) or the median (25th and 75th percentiles, interquartile range (IQR)). Friedman Repeated Measures Analyses of Variance on Ranks were performed to analyze changes in the aggregometric measurements and conventional laboratory data. Specific post-hoc tests were performed, if statistically significant differences were detected. Groups were compared using *t*-test or the Mann–Whitney *U* test as appropriate. Bonfferoni–Holm’s method was applied to adjust for multiple testing. Linear regression analysis was performed for ADP-, AA-, and TRAP-6-induced platelet aggregation as depending variable and paO2 measured in the oxygenator as determinant variable.

The statistical analyses were performed using Statistica 13.3 (Tibco Software Inc., Palo Alto, CA, USA).

## 3. Results

### 3.1. Population

From July 2014 to June 2015, a total of 20 consecutive patients with the need for vvECMO support were enrolled to the study. The sociodemographic and clinical characteristics of the patients are presented in Table 1. The flow of patients is shown in Figure 1, respectively.

During the observational period of seven days, seven patients showed signs of mild bleeding, one patient suffered from a severe bleeding complication (intracerebral bleeding) on the second day of ECMO treatment. This patient had a known cerebral ischemia on enrollment and ECMO treatment was initiated in awareness of the increased risk for an intracerebral bleeding event. Two patients out of 20 required an oxygenator or ECMO circuit change respectively due to thromboembolic complications. Three patients out of 20 received a transfusion of platelet concentrates during the observational period.

### 3.2. Laboratory Data

The median platelet count showed a significant drop 6 h after the implementation of ECMO therapy and displayed a declining trend during the remaining observational period with a further significant drop at T4 when compared to T0 (Figure 2). Leukocyte count and hemoglobin showed a significant reduction at T0 as well (Table 2). The remaining laboratory data is presented in Table 2.

### 3.3. MEA Data

The median AU was already lower than the normal reference range given by the manufacturer at T-1 in all three platelet function tests performed. Only patients that were placed on ECMO at the ARDS center could receive MEA measurements at T-1, because the method was not available at the referral hospitals and blood samples for MEA need to be processed quickly. Platelet aggregation after the stimulation with TRAP-6 showed a significant drop 6 h after the implementation of ECMO therapy (90 AU (59/151) vs. 75 AU (49/103) (*p* = 0.007)) and a spontaneous recovery with a significant increase two days after the initiation of ECMO (75 AU (49/103) (T0) vs. 111 AU (48/136) (T2) (*p* = 0.004)) (Figure 3). Both ASPI and ADPtest displayed a similar course with a drop in AU 6 h after the initiation of ECMO therapy and a spontaneous recovery within the first two days of observation, but without significant differences between the time points (Figure 3). There was no difference in MEA results between patients with and without signs of bleeding during the observational period.

Linear regression analysis showed a significant correlation between platelet function after the stimulation with ADP and the paO_2_ in the oxygenator at T4 (*p* = 0.015; *r*^2^ = 0.486). This correlation was not evident at any other timepoint.

## 4. Discussion

This study evaluated the effects of vvECMO therapy on platelet function over time and revealed a significant reduction in TRAP-6 induced platelet aggregation 6 h after the start of ECMO therapy and a spontaneous significant increase in TRAP-6 induced platelet aggregation two days after the implementation of ECMO. Platelet aggregation induced by AA and ADP appeared to follow the same trend. Additionally, we observed an inverse relationship between paO_2_ in the oxygenator and ADP-induced platelet aggregation at T4. 

In accordance with others [7,17,18], the platelet count dropped significantly 6 h after the beginning of ECMO therapy with a further decreasing trend during the complete observational period and a significant decrease after seven days of ECMO therapy. 

Reduced platelet aggregability during ECMO therapy, as observed in the present study, has been described with different methods [19,20,21]. In general, TRAP-6 induced platelet aggregation is considered to be too insensitive to reflect impairment of platelet function during ECMO therapy [20]. Our results demonstrate that the profound changes in homeostasis after the initiation of an extracorporeal circulation even influence this supposedly insensitive test for a short period of time. Although a correlation with an increased risk for hemorrhagic events cannot be answered by our data, it might be useful to monitor TRAP-6 induced platelet aggregation in the early phase of ECMO therapy to identify patients with an overall reduced platelet function.

In contrast to the continuously decreasing platelet count, platelet function fully recovered and exceeded baseline values during the remaining observational period after two days of extracorporeal circulation. This suggests that a routine monitoring of TRAP-6-induced platelet aggregation in the later stages of extracorporeal circulation does not seem useful. On the contrary results like this could even lead to an underestimation of platelet function impairment during longer-term ECMO therapy, that undoubtedly exists, since they might suggest “normality” to a less experienced user [21]. 

A potential reason for the higher values in TRAP-6-induced platelet aggregation after day two, especially since accompanied by a similar trend in AA- and ADP-induced platelet aggregation, might be a decreasing effect of factors like the initial hemodilution under extracorporeal circulation. Furthermore, MEA, as a whole blood method, is affected not only by platelet number and function but also by plasma, containing pro- as well as anticoagulatory factors [22]. Especially the latter can interfere with activation of the thrombin receptor which is critical for TRAP-induced aggregation. Since platelet aggregation after stimulation with TRAP-6 is rather insensitive, a normalization of the AU under ECMO therapy might partly reflect an improvement of the underlying disease due to intensive care therapy in general. The influence of critical illness on platelet count as well as on platelet function has been widely recognized [23,24,25,26]. Davies et al. demonstrated an influence of severe sepsis and septic shock on whole blood aggregometry [27]. Noticeably, more than 50% of the patients in the present study displayed aggregometric measurements below the normal reference range before the initiation of ECMO therapy. This might indicate that the reduced platelet aggregability seen during ECMO therapy could partly be due to the negative influence of the severe underlying disease or comorbidities on platelet function. 

To our knowledge, no studies have evaluated platelet function using MEA in patients suffering from severe ARDS without ECMO therapy so far. Passmore et al. observed an increased collagen-induced platelet aggregation during the first 24 h of extracorporeal circulation in sheep after smoke inhalation-induced ARDS [13]. In contrast, another study using the same ARDS model did not find significant changes during the first 24 h of extracorporeal circulation [28]. The authors observed a desensitization of platelets to ADP-induced aggregation caused by ECMO-induced hyperoxemia, which was not present in hyperoxemia without ECMO. In the present study, we detected a correlation between ADP-induced platelet aggregation and paO_2_ at T4. It can only be speculated why this effect was not present in the earlier phases of extracorporeal circulation. Although data from animal modes evaluating MEA are questionable, since it is not fully elucidated if ovine platelets react in the same way as human platelets [29,30], there are possible negative effects of oxygen respectively hyperoxemia [31,32,33]—which may occur during ECMO therapy—on platelet function. This would be an interesting aspect for further evaluation of hemostatic changes during ECMO therapy. On the other hand, the observed correlation between ADP-induced platelet aggregation and hyperoxemia might also suggest that a less impaired ADP-induced platelet function during ECMO therapy could be associated with a poorer oxygenator state due to microembolisations, hence resulting in lower paO_2_ values. 

Our study has some limitations. First, one patient, whose aggregometric measurements were included in the analysis, received platelet concentrates after T2. This was unavoidable due to a scheduled surgical intervention but might have affected the further MEA results. Second, it would have been of clinical interest to correlate aggregometric measurements with hemorrhagic or thromboembolic events to evaluate MEA as a potential predictor. This was not possible due to the small size of the cohort. Third, the use of MEA is limited by platelet count. This inevitably causes a certain bias due to a selection of patients with higher platelet counts in studies evaluating MEA. We tried to minimize this influence of platelet count, since patients with low platelet counts represent the reality in ECMO therapy. Nevertheless, no aggregometric measurements were included, if the platelet count dropped below the selected cut-off. This might have led to a selection of patients for the evaluation of MEA, whose platelets were less affected by extracorporeal circulation over time in the present study as well. Additionally, this practice reduced the number of patients, that could be evaluated with regard to MEA at different time points, including T0.

The dependency on platelet count represents a general problem for the routine use of MEA in platelet function monitoring during ECMO therapy, since low platelet count is one of its most common side effects. Undoubtedly, during severe hemorrhage, MEA will remain a valuable part of established algorithms for the management of hemotherapy during extracorporeal circulation. For an implementation into routine monitoring, the frequently occurring low platelet counts will be an impediment hard to overcome.

## 5. Conclusions

Platelet function assessed by multiple electrode aggregometry in patients with severe ARDS is already impaired before the initiation of extracorporeal circulation. As a clinical implication, platelet function should be assessed before initiation of ECMO and corrected if necessary. Although the start of ECMO therapy led to a further significant decrease in TRAP-6-induced platelet aggregation after 6 h with AA- and ADP-induced platelet aggregation following the same trend, all measurements had recovered to baseline values on day two. Therefore, an additional monitoring of TRAP-6-induced platelet aggregation may be considered in the early stages of ECMO therapy to identify patients with a markedly reduced platelet function. In the later stages, MEA and especially TRAP-6-induced platelet aggregation might not adequately reflect the changes in platelet function during extracorporeal circulation. 

## Figures and Tables

**Figure 1 jcm-08-01056-f001:**
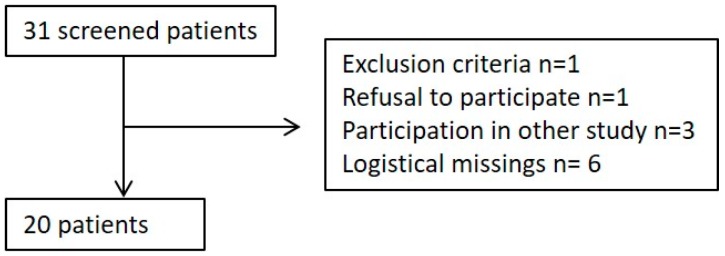
Flow of Patients during the screening process.

**Figure 2 jcm-08-01056-f002:**
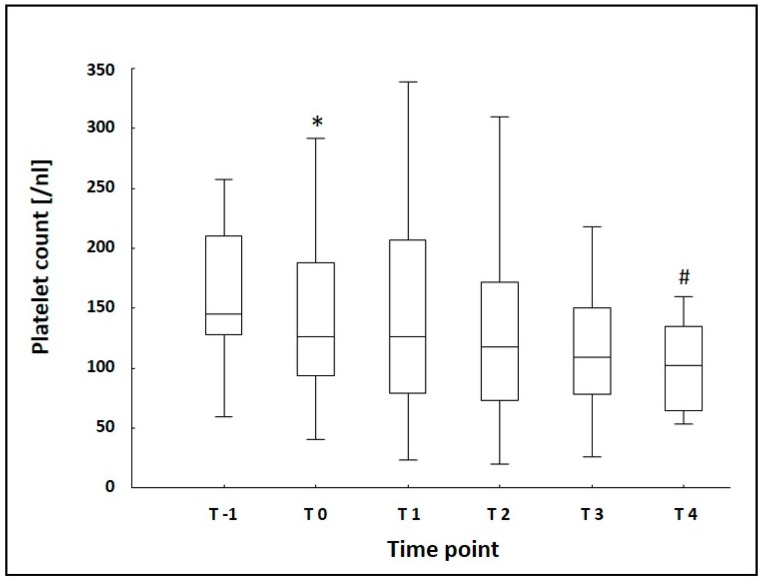
Platelet Count measured at baseline (T-1) and five times during the observational period of seven days; * *p* = 0.003 indicating significant difference between T-1 and T0, # *p* = 0.005 indicating significant difference between T0 and T4; blood samples were collected before (T-1), 6 h (T0), 1 day (T1), 2 days (T2), 3 days (T3), and 7 days (T4) after the start of ECMO therapy; boxes represent the interquartile range (25%–75%), whiskers represent the non-outlier range.

**Figure 3 jcm-08-01056-f003:**
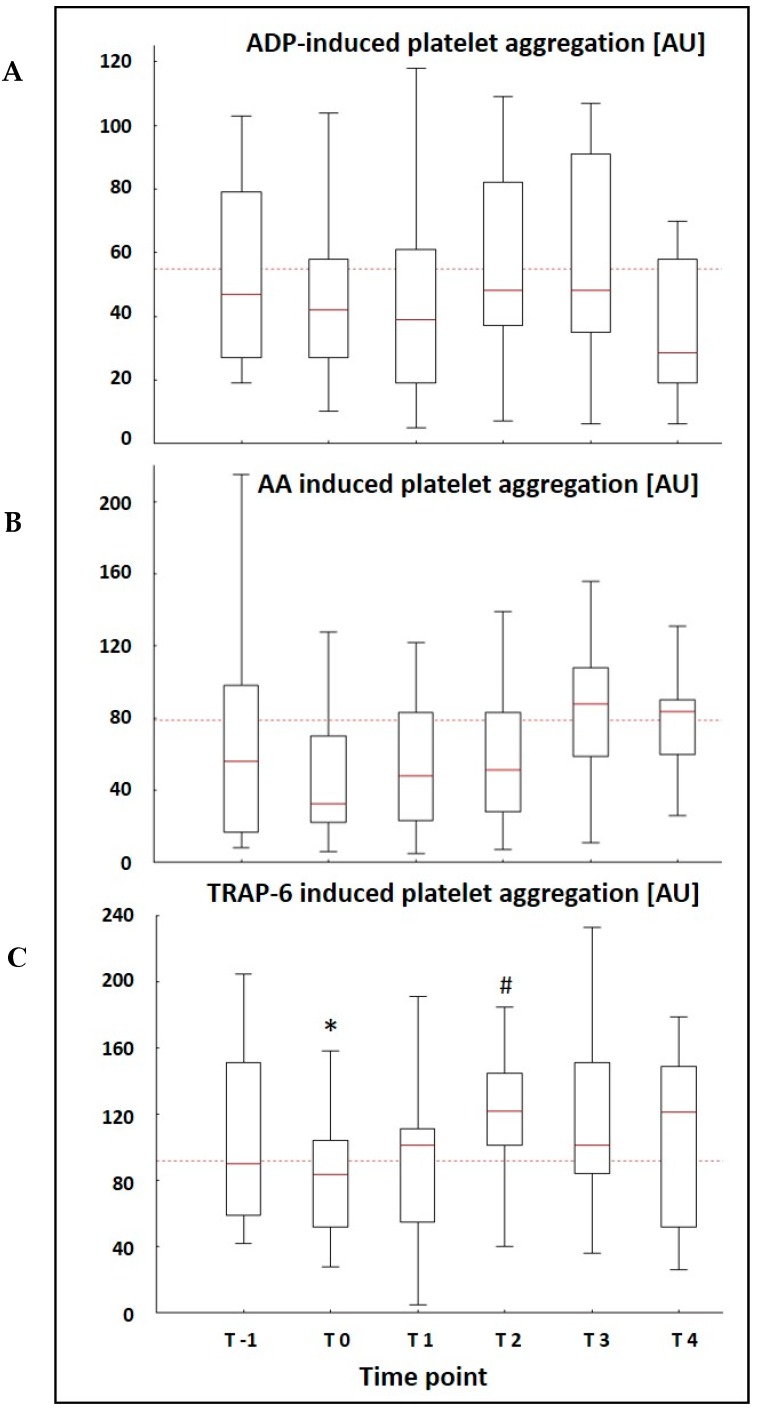
Platelet Aggregation assessed by MEA after stimulation with (**A**) ADP, (**B**) AA, and (**C**) TRAP-6 at baseline (T-1) and five times during the observational period of seven days; * *p* = 0.007 vs. T-1; # *p* = 0.004 vs. T0; dotted line is indicating the threshold to normal reference range; blood samples for platelet aggregometry were collected before (T-1) (*n* = 11), 6 h (T0) (*n* = 18), 1 day (T1) (*n* = 17), 2 days (T2) (*n* = 15), 3 days (T3) (*n* = 15), and 7 days (T4) (*n* = 11) after the start of ECMO therapy; boxes represent the interquartile range (25%–75%), whiskers represent the non-outlier range.

**Table 1 jcm-08-01056-t001:** Patient characteristics at baseline.

**Age** [years, median (IQR)]	57 (53/64)
**Gender** [Male, *n* (%)]	13 (65%)
**BMI** [kg/m^2^, median (IQR)]	30.7 (26.2/36.9)
**SAPS II** [median (IQR)]	40 (29/55)
**Reason for ARDS** [*n*]	1. Pneumonia *n* = 8 2. Aspiration *n* = 2 3. Postoperative ARDS *n* = 3 4. Smoke inhalation *n* = 1 5. Thorax trauma *n* = 1 6. Internal disease *n* = 5
**Duration of ECMO-Therapy** [days, median (IQR)]	9.5 (7/13.5)
**Duration of ICU-Therapy at the ARDS Center** [days, median (IQR)]	14.5 (11.2/35.6)
**Total Duration of ICU-Therapy** [days, median (IQR)]	19 (13.5/51)
**28-day Mortality** [*n* (%)]	9 (45%)
**ICU Mortality** [*n* (%)]	11 (55%)
**Hospital Mortality** [*n* (%)]	12 (60%)

**Table 2 jcm-08-01056-t002:** Laboratory Data.

Parameter	T-1 (*n* = 13)	T0 (*n* = 20)	T1 (*n* = 19)	T2 (*n* = 18)	T3 (*n* = 18)	T4 (*n* = 16)
**Hemoglobin [g/dL]**	10.1 (8.6/13)	8.9 * (8.3/10.1)	9.2 (8.4/10.1)	8.7 (8.3/9.5)	8.6 (8.3/9.1)	8.7 (8.2/9.7)
**Platelet Count [/nL]**	145 (128/210)	126 * (94/188)	126 (79/207)	118 (73/172)	109 (78/150)	102 ^#^ (65/135)
**Fibrinogen [mg/dL]**	465 (366/584)	433 (307/632)	426 (275/567)	434 (257/539)	406 (233/602)	329 (223/681)
**PT [%]**	82 (72/88)	74 (63/87)	78 (61/90)	79 (65/93)	78 (71/95)	77 (64/90)
**aPTT [s]**	32 (29/35)	32 (29/41)	36 (30/42)	35 (30/42)	34 (30/40)	37 (36/41)
**Leukocyte Count [/nL]**	16.7 (8.7/21.3)	12.2 * (6.7/16.2)	10.4 (8.2/15.9)	11.1 (7.5/14.5)	10.5 (8.1/14.7)	13.5 (8.4/16.8)

Data are given as median with interquartile range, * indicating *p* < 0.006 when compared to baseline at T-1; ^#^ indicating *p* = 0.0056 when compared to T0; PT, prothrombin time; aPTT, activated partial thromboplastin time.

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
