# Peer review of "Platelet Function Disturbance During Veno-Venous ECMO in ARDS Patients Assessed by Multiple Electrode Aggregometry—A Prospective, Observational Cohort Study"

_jcm, 2019, doi:10.3390/jcm8071056_

Reviewer 1 Report

This is a good work about TRAP-6-induced platelet aggregation after VV ECMO use. The author clearly presented the results about decreasing AA- and ADP-induced platelet aggregation after VV ECMO use for 6 hours.

Therefore, the author suggested monitoring TRAP-6-induced platelet aggregation after ECMO use. The author also provided hints about less impaired ADP-induced platelet function during ECMO therapy could be associated with a poorer oxygenator state due to microembolization, hence resulting in lower paO2 values.

I only suggested the author may consider shortening discussion and conclusion. This discussion is too long to understand the important findings of this work.

Author Response

Dear reviewer herewith we address your following comment.

Reviewer 1

This is a good work about TRAP-6-induced platelet aggregation after VV ECMO use. The author clearly presented the results about decreasing AA- and ADP-induced platelet aggregation after VV ECMO use for 6 hours.

Therefore, the author suggested monitoring TRAP-6-induced platelet aggregation after ECMO use. The author also provided hints about less impaired ADP-induced platelet function during ECMO therapy could be associated with a poorer oxygenator state due to microembolization, hence resulting in lower paO2 values.

 1.     I only suggested the author may consider shortening discussion and conclusion. This discussion is too long to understand the important findings of this work

 Comment and Changes: Following the reviewers comment, we shortened and focused the discussion. As a result the discussion length is reduced from  1.396 to 1006 words.  It is our impression that the manuscript improved from this revision. But if the reviewer does not agree with the new version, we are willing to further revise the discussion.

We included a red marked changes version, that points out the changes based on your comment.

Thank you very much for taking your time to review our article.

Kind regards

Onnen Mörer

Reviewer 2 Report

Dear authors,  I think this paper is suitable for publication after minor revisions of English language and style: please also check again your manuscript for typo and grammar errors. I suggest to better draw figure 1 wich is out of size in respect to other figures.

Author Response

Dear Reviewer,

we addressed the following comment:

Reviewer 2

1.     Dear authors,  I think this paper is suitable for publication after minor revisions of English language and style: please also check again your manuscript for typo and grammar errors.

Comment and Changes: We revised the manuscript accordingly.

2.     I suggest to better draw figure 1 which is out of size in respect to other figures.

Comment and Changes: We revised the manuscript accordingly

With kind regards

Onnen Moerer

Reviewer 3 Report

Authors of this study address an interesting question of how vvECMO therapy impacts platelet function. However, the results and discussion are not convincing enough. In fact, the time-dependent platelet aggregation does not show any significant change over time. The study overall seems incomplete and could have used additional or alternative methods to answer the question. 

Author Response

Dear Reviewer,

we herewith I like to revere to you comment:

Reviewer 3

Authors of this study address an interesting question of how vvECMO therapy impacts platelet function. However, the results and discussion are not convincing enough. In fact, the time-dependent platelet aggregation does not show any significant change over time. The study overall seems incomplete and could have used additional or alternative methods to answer the question

Comment:

We thank the reviewer for taking his time to read our manuscript and for his valuable comments.

We cannot change the nature of our results and the design of the study. However, following the other reviewers comments we revised our discussion which in our opinion  considerably improved our manuscript. In part this also addresses the critique of the third reviewer.

With kind regards

Onnen Moerer

Reviewer 4 Report

Dr. Wand and Co-workers present in this manuscript data about platelet function during veno-venous ECMO. In a prospective observational study of patients in ARDS the authors examined platelet function using multiple electrode aggregometry. In this work platelet aggregometry decreased during the first day of ECMO treatment and recovered until day seven. The presented data are very interesting and platelet function under ECMO is usually impaired. A sufficient diagnosis of platelet dysfunction is therefore very important in ECMO therapy.

Concerns:

1.       Were patients excluded if age was below 18 in this study?

2.       Page 5; Table 1: Duration of ICU-therapy: there are two values presented. Which one is the correct one?

3.       Page5; ll.145-150: the authors state that seven patients showed signs of mild bleeding. In the text only six patients are mentioned. Please clarify.

4.       Is there a reason that the platelet aggregometry values are lower than the treshold to normal reference in the beginning of ECMO-therapy? The manuscript can be improved if a discussion about this fact is added. Is there a control group of patients without ECMO in your ICU to exclude that there is systematic error?

5.       The paper should highlight more the specific new character of the presented data. At the moment the exclusive new insights are not really highlighted. I would encourage the authors to do so. 

6.       Page 5, Figure 2:

1.       Asterixis and hashtag are missing in the figure.

2.       In which format are boxplots presented? Please state in the legend. This is also valid for Page 7; Figure 3.

Author Response

Dear Reviewer,

we like to respond to your comments. I think that we were able to address all points properly but we are of course willing to further adapt our manuscript if required.

Reviewer 4

Dr. Wand and Co-workers present in this manuscript data about platelet function during veno-venous ECMO. In a prospective observational study of patients in ARDS the authors examined platelet function using multiple electrode aggregometry. In this work platelet aggregometry decreased during the first day of ECMO treatment and recovered until day seven. The presented data are very interesting and platelet function under ECMO is usually impaired. A sufficient diagnosis of platelet dysfunction is therefore very important in ECMO therapy.

Concerns:

1.       Were patients excluded if age was below 18 in this study?

Comment and Changes: Age below 18 was an exclusion criterion, which is now placed as an information in the methods section. 

2.       Page 5; Table 1: Duration of ICU-therapy: there are two values presented. Which one is the correct one?

Comment and Changes: We are thankful that the reviewer saw this mistake. The first value represents the median duration of ICU-therapy at the ARDS center, the second values represent the total duration of ICU-therapy including the days at the referral hospital. We revised table 1 accordingly(see revision).

3.       Page5; ll.145-150: the authors state that seven patients showed signs of mild bleeding. In the text only six patients are mentioned. Please clarify.

Comment and Changes: We thank the reviewer for this comment, since it suggests that we have been not precise enough in the characterization of our patients. This might have led to the misunderstanding, that the patients mentioned in p.5; ll. 145-150 were a closer characterization of the seven patients suffering from mild bleeding complications. In fact, seven patients out of 20  suffered from mild signs of bleeding, one patient out of 20 had a severe complication (intracerebral bleeding), two patients out of 20 suffered from thromboembolic complications. We revised the section of the manuscript to ensure for more precision.

4.       Is there a reason that the platelet aggregometry values are lower than the treshold to normal reference in the beginning of ECMO-therapy? The manuscript can be improved if a discussion about this fact is added. Is there a control group of patients without ECMO in your ICU to exclude that there is systematic error?

 Comment and Changes: Our interpretation of these results is, that platelet function is already impaired in patients who are placed on ECMO therapy due to the cause of the disease prior to initiation. These patients are in multi organ dysfunction, have Sepsis and/or a state after major surgery. Thus in contrary to our expectation we have to correct our hypothesis that ECMO and the extracorporeal circuit is the major driver responsible for the platelet dysfunction, but also the clinical situation at onset of ECMO. For this reason, an assessment prior to cannulation might support therapy planning during the following course. In accordance with this argumentation a paragraph has been added to the discussion section.    

 5.       The paper should highlight more the specific new character of the presented data. At the moment the exclusive new insights are not really highlighted. I would encourage the authors to do so. 

Comment and Changes: We hope that the changes made to the discussion are sufficient to allow a better distinction of the new insights provided by our data. (see discussion section of the revised manuscript)

6.       Page 5, Figure 2:

  Asterixis and hashtag are missing in the figure.

Comment and Changes: 

Thank you very much for this comment.We corrected Figure 2.                                                                                                                       

2.       In which format are boxplots presented? Please state in the legend. This is also valid for Page 7; Figure

Comment and Changes:  We added the information into the Figure legend that explains the variables

 Wit kind regards

Onnen Moerer 

Round  2

Reviewer 1 Report

The revised manuscript improved much. It is a good work for critical care medicine.

Author Response

Dear Reviewer Nr. 1,

thank your for your comment and review.

Best regads

Onnen Moerer

Reviewer 3 Report

The discussion has definitely improved. Considering the nature of these results, this article is good in its present form.

Author Response

Dear Reviewer 3,

thank you for you work as a reviewer of our manuscript.

Best regards

Onnen Moerer

Reviewer 4 Report

The authors sufficiently adressed all concerns in the revised manuscript.

There remains just a minor question:

P.9. l.241: ...or comorbidities on platelet function. XXX Sepsispaper

That is an unusal citation style.

Author Response

To Reviewer 4,

P.9. l.241: ...or comorbidities on platelet function. XXX Sepsispaper

That is an unusal citation style#

Comment: We thank the reviewer for his comment and apologize for the additional work. "xxx Sepsispaper"  was a place holder before adding the paragraph before.

Changes: Deleated without additional changes.

Best regards

Onnen Moerer